# Swell hindcast statistics for the Baltic Sea

Jan-Victor Björkqvist[1,2,3], Siim Pärt[2], Victor Alari[2], Sander Rikka[2], Elisa Lindgren[3], and Laura Tuomi[3]

[1]Norwegian Meteorological Institute, Allégaten 70, 5007, Bergen, Norway
[2]Department of Marine Systems, Tallinn University of Technology, Akadeemia tee 15a, 12611, Tallinn, Estonia
[3]Finnish Meteorological Institute, P.O. Box 503, 00101 Helsinki, Finland

**Correspondence:** Jan-Victor Björkqvist (jan-victor.bjorkqvist@met.no)

**Abstract.** The classic characterisation of swell as regular, almost monochromatic, wave trains doesn't necessarily accurately describe swell in water bodies shielded from the oceanic wave climate. In such enclosed areas the locally generated swell waves still contribute to processes at the air and seabed interfaces, and their presence can be quantified by partitioning wave components based on their speed relative to the wind. We present swell statistics for the semi-enclosed Baltic Sea using 20 years of swell partitioned model data. The swell significant wave height was mostly under 2 m, and in the winter (DJF) the mean significant swell height was typically less than 0.4 m; higher swell was found at limited nearshore areas. Swell waves were typically short (under 5 s), with mean periods over 8 s being rare. In open-sea areas the average ratio of swell energy (to total energy) was mostly below 0.4 – significantly less than in World Ocean. Certain coastal areas were swell dominated over half the times, mostly because of weak winds ($U < 5 \, \mathrm{ms}^{-1}$) rather than high swell heights. Swell dominated events with a swell height over 1 m typically lasted under 10 h. A cross-correlation analysis indicates that swell in the open sea is mostly generated from local wind-sea when wind decays (dominant time lag roughly 15 h). Near the coast, however, the results suggest that the swell is partially detached from the local wind-waves, although not necessarily from the weather system that generates them.

## 1 Introduction

Sea surface waves are generated by the wind blowing over the water. They then travel to distant areas while continuing to interact with their environment even after the wind has ceased, thus becoming swell. Once reaching the shore the swell releases its energy through breaking and other wave–seabed interactions. Swell waves therefore serve as a spatial transfer mechanism – and a time-buffer – of the kinetic energy and momentum contained in the atmosphere, thus offering an additional opportunity to harvest this renewable energy. In air–sea interaction studies, again, this de-correlation with the local wind regime is an unwanted property, since swell waves taint the measurements with information of past wind and wave conditions from outside the study area, and can even cause an upward momentum flux from the swell to the atmosphere (Semedo et al., 2009; Kahma et al., 2016). Swell can also affect oil transport within the mixed layer by – depending on swell–wind angle – either enhancing or suppressing the vertical eddy diffusivity (Chen et al., 2016).

Swell waves are common (Ardhuin et al., 2009), and the longest swell waves generated in the World Ocean can travel across entire ocean basins before reaching the shoreline (Alves, 2006). The global swell climate has been studied using model

simulations (Semedo et al., 2011; Fan et al., 2014; Amores and Marcos, 2020), but our knowledge about the propagation of swell is still limited by the scarcity of measurements (Babanin et al., 2019). An especially heavy swell climate is found on the Pacific Ocean coastlines (Yang et al., 2019), but swell is persistent also at the coasts of the Atlantic Ocean (Vettor et al., 2013; Semedo et al., 2015). However, there are also coastal areas of the World Ocean that are well sheltered from the dominant
ocean swell, with the swell climate being more characterized by locally generated swell waves (Hanley et al., 2010; Qian et al., 2020).

Conceptually swell is often thought of as regular and long-crested waves that are almost monochromatic and highly directional (e.g. Holthuijsen, 2007, page 47). In the World Ocean this characterization is apt (Barber and Ursell, 1948), and might also coincide with a layperson's view on swell. Nonetheless, in wave measurements and wave models swell needs to be quan-
tified, and the definition of swell is typically (loosely speaking) taken as waves outrunning the wind (e.g. Bidlot, 2001). As a result, swell is taken simply as a selection of wave components that fulfil this criteria. While the conceptual regular swell waves also fulfil the quantitative criteria, the opposite is not necessarily true – although this is subjective. Be that as it may, the quantitative criteria is still well motivated from a standpoint of air–sea interaction, including wind-wave growth (e.g. Komen et al., 1984; Kahma et al., 2016).

In enclosed or semi-enclosed seas the swell climate is completely detached from the global swell conditions (Berkun, 2007; Van Vledder and Akpınar, 2016; Divinsky and Kosyan, 2018). One such semi-enclosed sea is the Baltic Sea, which is only connected to the ocean through the narrow and shallow Danish straits, through which no significant amount of swell can propagate.The small size of the Baltic Sea naturally limits the severeness of its swell climate compared to the oceans, and waves classified as swell might often not coincide with the above mentioned conceptual definition. Even locally generated
swell has still been found to be an issue in e.g. air–sea interaction studies (Smedman et al., 1999; Carlsson et al., 2009). A refined understanding of the Baltic Sea swell climate is also of interest because of the heavy marine traffic and the consequent risk of oil pollution (HELCOM, 2018).

Up until now, the swell climate of the Baltic Sea has not been studied by any dedicated model simulations, although it has been quantified as a byproduct of a 10 km hindcast covering a larger domain (Semedo et al., 2014). Our study quantifies swell
in the Baltic Sea using 20 years of data from a Baltic Sea specific 1 nmi wave product. The paper is structured as follows: Section 2 presents the data and definitions, Section 3 presents statistics of the swell height, period, and direction, quantifies the prevalence of swell, and investigates the cross-correlation structure between wind-sea and swell waves. Section 4 is dedicated to discussing our results, and Section 5 ends the paper by summarising our conclusions.

## 2 Materials and Methods

### 2.1 Model data

The wave model data originate from the Copernicus Marine Environment Monitoring Service's (CMEMS) hindcast product BALTICSEA_REANALYSIS_WAV_003_015. This simulation was made by the The Baltic Monitoring Forecasting Centres (BAL MFC) Production Unit at Finnish Meteorological Institute (FMI) for the CMEMS service (Lindgren et al., 2020) and

was compiled using the wave model WAM (WAMDIG, 1988; Komen et al., 1994), with some modifications and additions
to account for specific features of the Baltic Sea (Tuomi et al., 2011, 2014). The WAM (Cycle 4.6.2) model domain covers
53° N–66° N, 9° E–30° E with a 1 nmi (1.85 km) resolution, and the wave spectra has 35 logarithmically spaced frequencies
(0.0418–1.067 Hz) and 24 directions with 15 degree intervals.

The atmospheric forcing for the model is a European Centre for Medium-Range Weather Forecasts (ECMWF) Reanalysis
5th Generation (ERA5) reanalysis for wind, and has a 0.28 degree (approximately 31 km) resolution (Hersbach et al., 2020).
Boundary spectra at the open boundary of Skagerrak originated from the ERA5 wave reanalysis, mainly affecting the wave
conditions near Skagerrak and Kattegat, which are outside our focus area. The northern parts of the Baltic Sea freeze annu-
ally, and the seasonal ice cover was therefore accounted for by an ice mask that excluded wave model grid points from the
calculations if the ice concentration exceeded 30 %. The ice concentrations were taken from the gridded ice charts produced
by SMHI (Swedish Meteorological and Hydrological Institute) and FMI, and the digitized product had a resolution of 3 nmi
for 1993–2015, 2.4 nmi for 2015–2017, and 0.25 nmi for 2017–2018. The temporal resolution of the ice data varies. Between
1992 and 2005 the resolution is 3-4 days but from March 2005 onward the resolution is mostly 1 day.

The CMEMS wave product covers the period 1993–2018 and consists of hourly outputted wave parameters, including
partitioned primary and secondary swell parameters. For this study we used total swell parameters for the 20 year period
1999–2018. These total swell parameters are not available in the CMEMS database, but still originate from the same model
simulation.

The validation of the CMEMS product compared the simulated significant wave height and peak wave period to observations
from nine wave buoys, while the mean wave period ($T_{m_{02}}$) was compared to data from four of these buoys (Lindgren et al.,
2020). The validation reveals that the overall accuracy of the produce is good, with the mean significant wave height being
slightly too low in the open sea and slightly too high near the coast. Overall, the significant wave height had a bias and
root-mean-square error (RMSE) of -0.04 m and 0.24 m. The peak periods for higher significant wave heights are fairly well
reproduced in the product, but periods during low significant wave heights show an expected larger scatter. The overall bias
and RMSE for the peak period are -0.45 s and 1.09 s. The mean wave period is slightly overestimated in the product, with the
bias and RMSE over the whole domain being 0.26 s and 0.69 s.

## 2.2 Swell partitioning

The total significant wave height is defined as

$$H_s = H_{m_0} = \int\limits_{-\pi}^{\pi}\int\limits_{0}^{\infty} S(\omega,\theta)\,\mathrm{d}\omega\mathrm{d}\theta,$$ (1)

where $S(\omega,\theta)$ is the wave spectrum, $\omega$ is the angular frequency, and $\theta$ is the wave direction.

Further, the spectral swell partitioning used in WAM classifies energy as swell if the following criteria is fulfilled (Bidlot, 2001):

$$1.2 \left( \frac{28 u_*}{c} \right) \cos(\theta - \Phi) \leq 1, \tag{2}$$

where $u_*$ is the friction velocity, $c$ is the phase speed of the wave component and $\Phi$ is the wind direction. The swell spectrum, $S^{swell}(\omega, \theta)$, is therefore made up of the spectral bins $S(\omega, \theta)$ that fulfills the above criteria, being 0 elsewhere.

The swell significant wave height is defined by integrating the swell spectrum

$$H_s^{swell} = \int_{-\pi}^{\pi} \int_0^{\infty} S^{swell}(\omega, \theta) \, \mathrm{d}\omega \mathrm{d}\theta, \tag{3}$$

and the swell mean period is defined using the inverse moment

$$T_m^{swell} = 2\pi \frac{\int_{-\pi}^{\pi} \int_0^{\infty} \omega^{-1} S^{swell}(\omega, \theta) \, \mathrm{d}\omega \mathrm{d}\theta}{\int_{-\pi}^{\pi} \int_0^{\infty} S^{swell}(\omega, \theta) \, \mathrm{d}\omega \mathrm{d}\theta}. \tag{4}$$

Following Semedo et al. (2011) we define the swell energy weight as

$$W_S = \left( \frac{H_s^{swell}}{H_s} \right)^2. \tag{5}$$

This dimensionless variable takes values between 0 (no swell present) to 1 (all energy is classified as swell). We classify the sea state as swell dominated if $W_S > 0.5$, i.e. over half of the energy is considered swell.

The wind-sea significant wave height fulfills

$$H_s^{sea} = \sqrt{H_s^2 - (H_s^{swell})^2}. \tag{6}$$

## 2.3 Statistics

The seasonal ice cover of the Baltic Sea complicates the definition of wave statistics (Tuomi et al., 2011). When ice is present, two of the possible type of statistics are: ice-free statistics (Type F) and ice included statistics (Type I). In Type F the statistic (e.g. mean value) is calculated using only the times when the grid point is ice-free. In Type I the ice time is included in the calculations by assuming that the ice cover blocks the waves, i.e. by setting $H_s = 0$. We use Type I statistics for the significant swell height, $H_s^{swell}$. The swell energy weight, $W_S$, is only defined for $H_s > 0$, which is why we use Type F statistics for this parameter.

## 3  Results

### 3.1  Swell height

In the Baltic Proper the mean swell significant wave height ($H_s^{swell}$) was mostly below 0.4 m during the winter (DJF) (Fig. 1a). Small areas close to the coasts near Klaipėda, Kaliningrad, and Gotland showed larger values – up to ca 0.8 m. These coastal areas in the Baltic Proper also had the largest mean swell height (0.3–0.4 m) during the summer (JJA) (Fig. 1b).

In the smaller sub-basins the mean swell heights were heavily influenced by the seasonal ice cover. The Baltic Sea starts freezing from the Bay of Bothnia and the eastern Gulf of Finland, where the ice cover can last even until May (SMHI and FIMR, 1982). As a result, the mean swell height in the Gulf of Finland and Bay of Bothnia was actually lower for the winter season compared to the summer. In the Gulf of Riga and the Bothnian Sea the mean swell heights were similar for both the summer and the winter season.

Björkqvist et al. (2018) found that most over 7 m wave events takes place between November and January. These high waves turned into swell after the wind decayed, leading to the 99th percentiles of the significant swell height being around 2 m (winter) and below 1.4 m (summer) in larger parts of the Baltic Proper (Fig. 1c–d). The highest swell height in the northern Baltic Proper (5.5 m) occurred after the storm Rafael in December 2004, and this event shows a rapid re-classification of wind-sea energy to swell energy (Fig. 2).

The ice cover affected the 99th percentiles significantly less compared to the mean values. The 99th percentile for the winter months exceeded 1 m in the entire Bothnian Sea, with the lowest values found in the southeastern part of the basin (Fig. 1c). The mean heights exceeded 0.8 m also during the summer season (Fig. 1d). In the Bay of Bothnia and the Gulf of Finland the 99th percentiles were roughly similar both for the winter and summer season.

### 3.2  Swell prevalence

We quantified the prevalence of swell using the swell energy weight, $W_S$, which is the fraction of swell energy with respect to the total energy (Eq. 5). This parameter is undefined for the ice-time (see Sect. 2.3), and the statistics were therefore calculated for the ice-free time only (Type F).

During the winter (DJF) the mean swell weight was 0.2–0.3 in the larger parts of the Baltic Proper, being below 0.2 in large parts of the sub-basins (Fig. 3a). During the summer months (JJA) the mean swell weights were roughly 0.1 higher compared to the winter, exceeding 0.3 almost in the entire Baltic Sea (Fig. 3b) These are low numbers compared to the World Ocean, where the mean swell weight exceeds 0.5 (Semedo et al., 2011). Nevertheless, coastal sections with a mean swell weight over 0.5 were found in every sub-basin of the Baltic Sea, with the exception of Gulf of Riga. In the Baltic Proper the highest swell weights were along the eastern coastlines, which is expected since the prevailing southwesterly winds (Karagali et al., 2014) lead to an east–west asymmetry in the wave climate of the Baltic Proper (e.g. Tuomi et al., 2011; Björkqvist et al., 2018, and our Fig. 1c). Nonetheless, also a short coastal section in southeastern Gotland had mean swell weights exceeding 0.6.

In addition to the average swell weight, we also calculated the probability that the sea-state in any given location was swell dominated (defined as $W_S > 0.5$). During the winter (DJF) the sea state in larger parts of the Baltic sea had a 20–30 %

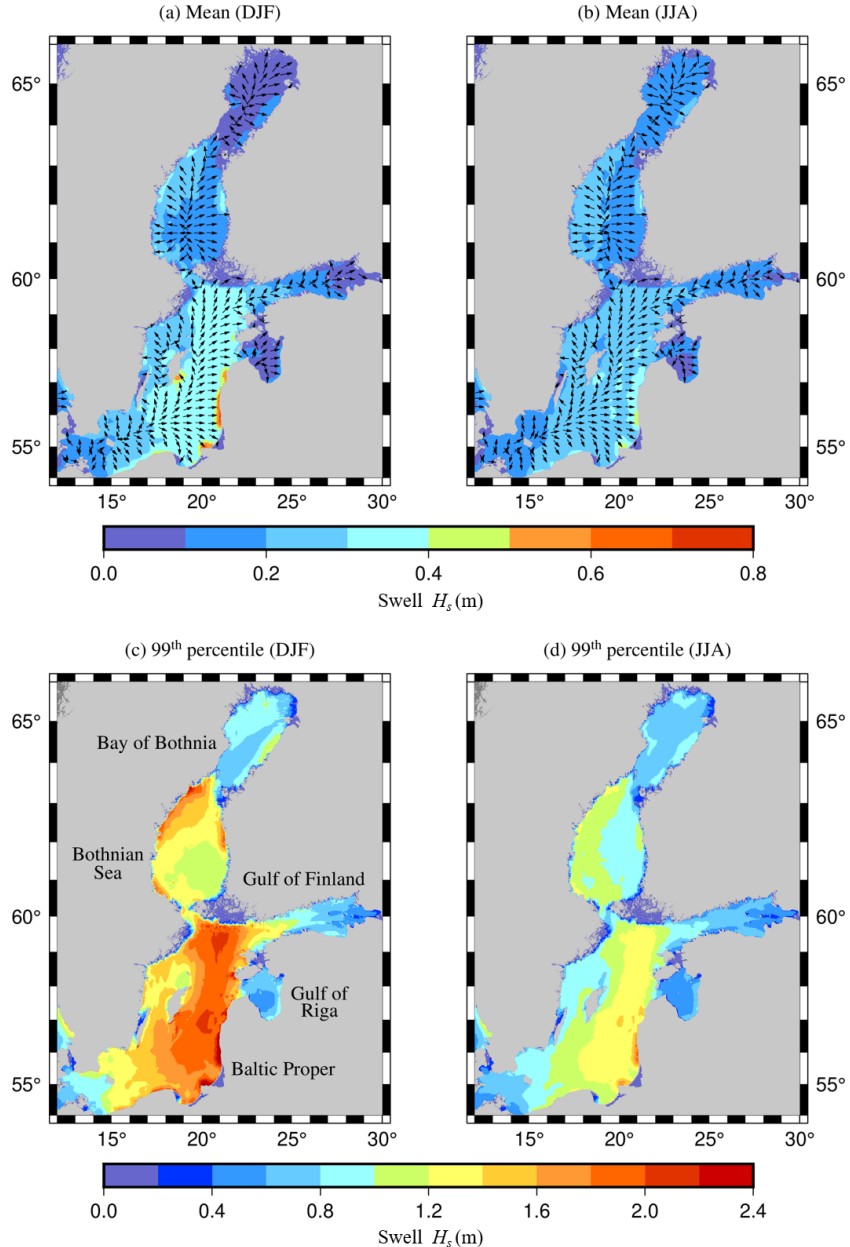

**Figure 1.** Mean values and 99[th] percentiles of swell significant wave height for winter (DJF) and summer (JJA). Note the different colour scales. The arrows signify the mean swell direction averaged over the seasons in the 20 year data. Ice-included statistics (Type I, Tuomi et al., 2011).

probability of being dominated by swell (Fig. 3c), which is low compared to 75–100 % in the World Ocean (Semedo et al., 2011) and the North Sea (ca. 40 % Semedo et al., 2015). During the summer (JJA) the Baltic Sea main basin had a 40–50

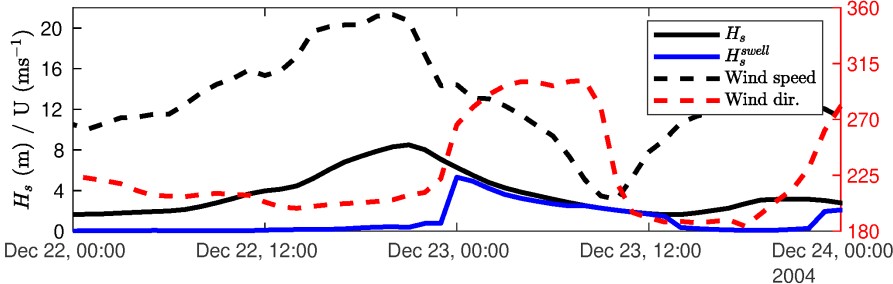

**Figure 2.** Highest swell case at the Northern Baltic Proper wave buoy (5.5 m), which took place during the storm Rafael.

% probability of being swell dominated (Fig. 3d). Such a general difference between the seasons was also identified for the North Sea and Norwegian Sea by (Semedo et al., 2015). The regions that were most often swell dominated (over 70 % of the times) were closely the same nearshore areas that had the highest mean swell weight. Longer coastal sections that were swell dominated more often than not could be found in all sub-basins except the Gulf of Riga.

We next focus on six locations from the largest sub-basins of the Baltic Sea. They are: Klaipėda, an area with high swell concentration in the southern Baltic Proper; Gotland, at the permanent wave buoy outside the land-based air–sea interaction tower at Östergarnsholm; Northern Baltic Proper (NBP), Gulf of Finland (GoF), and Bothnian Sea at the locations of FMI's operational wave buoys; and Kalajoki, outside a recreational sandy beach in the Bay of Bothnia. The locations are shown on the map in Fig. 3a.

The nearshore locations (Klaipėda and Kalajoki) were swell dominated around 70 % of the times, with the respective number being only 30–40 % for the open-sea locations (Table 1). The swell dominated cases were often characterized by a low wind speed (under 5 $\mathrm{ms}^{-1}$). For example: when the wind exceeded 5 $\mathrm{ms}^{-1}$, the sea state outside of Östergarnsholm, Gotland was swell dominated only 14 % of the times, and practically swell free ($W_S \leq 0.1$) 73 % of the times.

In an absolute sense, the swell height was almost always below 2 m at all locations, with heights over 1 m being rare outside the Baltic Proper (Table 1). High (over 1 m) dominant ($W_S > 0.5$) swell typically didn't persist for long (not shown). The median duration of such events was between 2–8 h at the six locations. High dominant swell was most prevalent at Klaipėda, where the longest case lasted 86 h.

### 3.3 Swell direction

The seasonally averaged mean swell directions did not reveal anything unexpected; the dominant direction in the southern Baltic Proper was from southwest, following the geometry of the basin up north. Waves were strongly refracted towards the coast, especially in the southern Baltic Sea where the waves refract into the bay of Gdansk. The swell direction was perpendicular to the coast at the exposed areas on the eastern coasts of the Baltic Proper. Although typical swell periods in the Baltic Sea are short (see Sec. 3.4), they are still, on average, longer than the wind-sea, and refracted in the relatively shallow coastal areas. In the Bothnian Sea the averaged swell directions were towards the coast with a clear divergence area in the

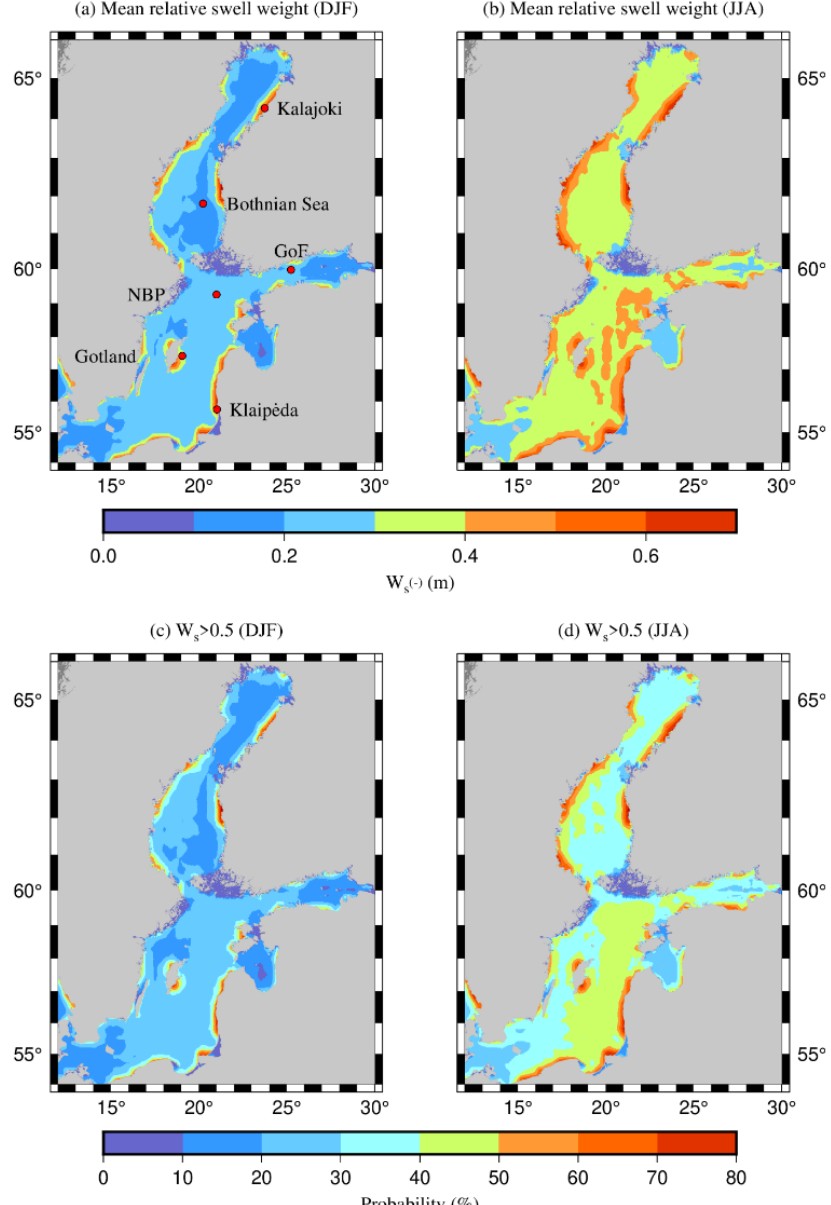

**Figure 3.** Swell energy weight, $W_S$: a) mean values, and b) probability of the sea state being swell dominated. Red dots show the locations of Table 1. Ice-free statistics (Type F, Tuomi et al., 2011).

middle of the basin – a pattern already identified by Semedo et al. (2014). There was no major differences in the directionality of swell between winter and summer.

**Table 1.** Exceedance probabilities of swell energy weight ($W_S$) and swell significant wave height ($H_s^{swell}$) at six locations (see Fig. 3a). Statistics are ice-free and ice-included respectively (Type F & I, Tuomi et al., 2011).

| | Klaipėda | Gotland | NBP | GoF | Bothnian Sea | Kalajoki |
|---|---|---|---|---|---|---|
| $W_S$ (-) | | Exceedance probability (%) | | | | |
| 0.10 | 81 | 60 | 50 | 47 | 39 | 86 |
| 0.20 | 79 | 57 | 46 | 42 | 36 | 83 |
| 0.30 | 77 | 53 | 42 | 38 | 34 | 81 |
| 0.40 | 74 | 49 | 38 | 33 | 31 | 78 |
| 0.50 | 69 | 43 | 34 | 28 | 28 | 73 |
| 0.60 | 62 | 36 | 29 | 24 | 24 | 66 |
| 0.70 | 53 | 29 | 24 | 20 | 20 | 55 |
| 0.80 | 42 | 21 | 18 | 15 | 16 | 41 |
| 0.90 | 28 | 14 | 12 | 10 | 10 | 26 |
| $W_S$ (-) | Exceedance probability when wind speed over 5 ms$^{-1}$ (%) | | | | | |
| 0.10 | 30 | 27 | 25 | 17 | 14 | 21 |
| 0.20 | 29 | 25 | 21 | 14 | 12 | 20 |
| 0.30 | 28 | 22 | 18 | 11 | 10 | 19 |
| 0.40 | 26 | 18 | 14 | 7 | 8 | 18 |
| 0.50 | 23 | 14 | 11 | 4 | 6 | 15 |
| 0.60 | 19 | 9 | 8 | 2 | 4 | 12 |
| 0.70 | 13 | 5 | 5 | 1 | 2 | 7 |
| 0.80 | 7 | 2 | 2 | 0 | 1 | 2 |
| 0.90 | 2 | 0 | 1 | 0 | 0 | 0 |
| $H_s^{swell}$ (m) | | Exceedance probability (%) | | | | |
| 0.25 | 59 | 46 | 42 | 24 | 24 | 30 |
| 0.50 | 38 | 22 | 22 | 8 | 9 | 12 |
| 0.75 | 25 | 9 | 12 | 2 | 3 | 4 |
| 1.00 | 16 | 4 | 6 | 0 | 1 | 2 |
| 1.25 | 10 | 1 | 3 | 0 | 0 | 0 |
| 1.50 | 6 | 1 | 2 | 0 | 0 | 0 |
| 1.75 | 3 | 0 | 1 | 0 | 0 | 0 |
| 2.00 | 2 | 0 | 0 | 0 | 0 | 0 |

The misalignment between the swell direction and the wind direction are roughly similar for all the six locations, namely small angles are most common (Fig. 4). The exception is Kalajoki (f) located near a sandy beach in the Bay of Bothnia. We surmise that the more erratic statistics at this point are a results of the ice-cover changing the fetch geometry; the data are thus essentially not from a single distribution. Wind–swell angles are also expected to be affected by the slanting fetch, especially in the Gulf of Finland where the wind–wave angle can be up to 50° even for wind-sea (Pettersson et al., 2010).

For the 99$^{th}$ percentile of swell heights the misalignment roughly follow that of the full data set at GoF and Bothnian Sea (Fig. 4 d & e), while the smallest angles are pronounced at Klaipėda and Kalajoki (a & f). For Gotland and NBP (b & c) the most probable misalignment is large, 60–100°; these large angles are indicative of sharp and sudden turns in the wind direction, as seen in e.g. Fig. 2. The smaller angles at Klaipėda and Kalajoki, again, suggest that the dominant mechanism for waves being classified as swell close to the coast might simply be the attenuation of the wind speed, although the distribution is surely site specific because of depth-induced refraction. The differences between the generation mechanisms of swell in the open sea and coastal areas will be further explored in Sec. 3.5.

## 3.4 Swell periods

A majority of the swell waves at the six locations had a mean period below 5 s (Fig. 5). In the Baltic Proper (Klaipėda, Gotland, and NBP) 67–80 % of the swell cases had a mean period under 5 s. In the GoF, Bothnian Sea, and Kalajoki almost all swell cases (82–91 %) had a mean swell period under 5 s. For all locations the swell significant wave height had a maximum value of 1.0–1.2 m when the period was under 5 s.

Long swell – with a mean period above 8 s – was almost non-existent in GoF, Bothnian Sea, and Kalajoki. Such long swell waves were rare also in the Baltic Proper, constituting only 1–2 % of all swell cases at Klaipėda, Gotland, and NBP.

## 3.5 Correlation of wind-sea and swell

The correlation of the wind-sea and swell significant wave heights were negative in the open sea, but positive for coastal areas (Fig. 6). The negative correlation of the open-sea areas were indicative of decaying winds turning existing wind-sea into swell in a zero-sum fashion. An example is the highest swell case at NBP during the storm Rafael (Fig. 2), where high waves were rapidly classified as swell when the wind turned as the cyclone passed. To study the correlation in more detail we determined the cross-correlation between swell and wind-sea heights (Fig. 7). The cross-correlation structure for open-sea locations (b–e) had maximum correlations of 0.2–0.3, which took place for 13–15 h time lags. These time lags correspond to the life cycle of waves generated by a larger scale weather pattern.

The positive correlation in the coastal areas suggests that the origin of the swell energy is different than in the open sea. Indeed, the cross-correlation functions at the two coastal locations were different compared to those from the open sea. The maximum peaks in the cross-correlation functions were stronger ($R = 0.5$–0.6), but took place for a shorter time lag (7–11 h). This indicates that the wind-sea in the coastal locations grows simultaneously as longer waves arrive from the open sea. These longer waves are generated by the same weather system, but because the wind speed decays towards the coast, they are classified as swell near the shoreline. The time lag is created by the different arrival times of a weather system (generating a

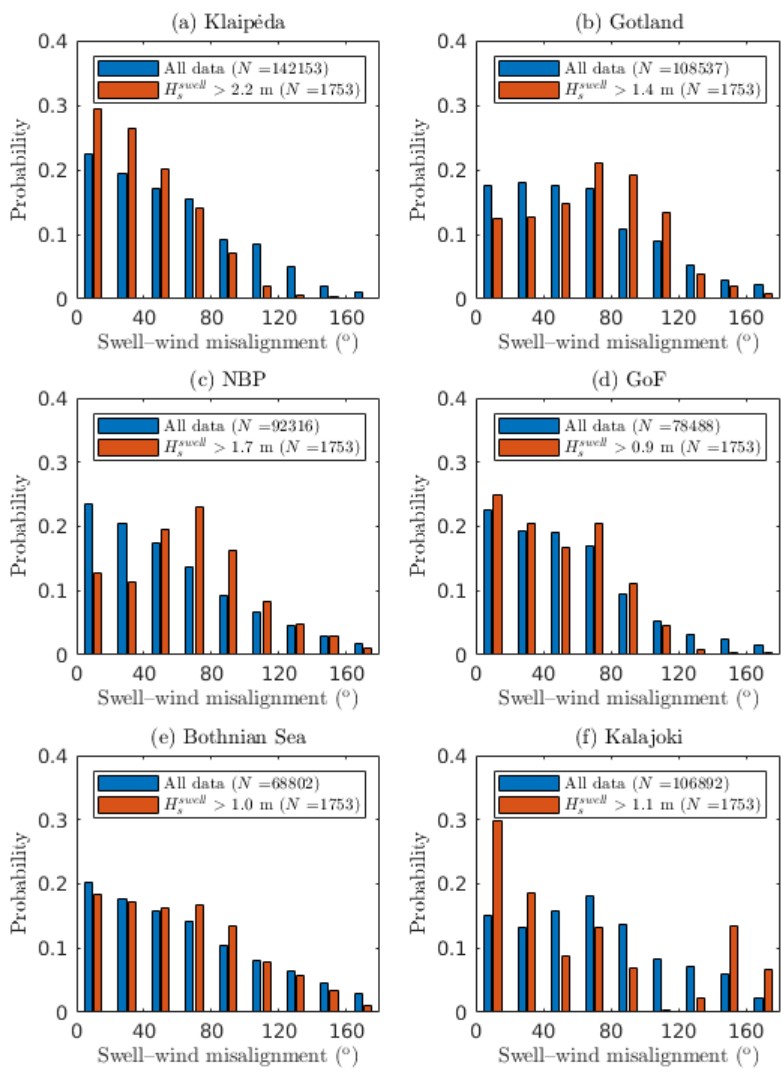

**Figure 4.** The misalignment between the swell direction and the wind direction for cases where the swell weight is $W_S > 0.05$. The red bars are instances where the significant swell height, $H_s^{swell}$, exceeds the 99[th] percentile. Ice-free statistics (Type F, Tuomi et al., 2011).

local wind-sea) and open-sea waves generated by that system. Nonetheless, coastal locations also inherit a similar correlation
structure to that of the open-sea waves during decaying winds, and the final connection between swell and wind-sea surely
depends on several factors.

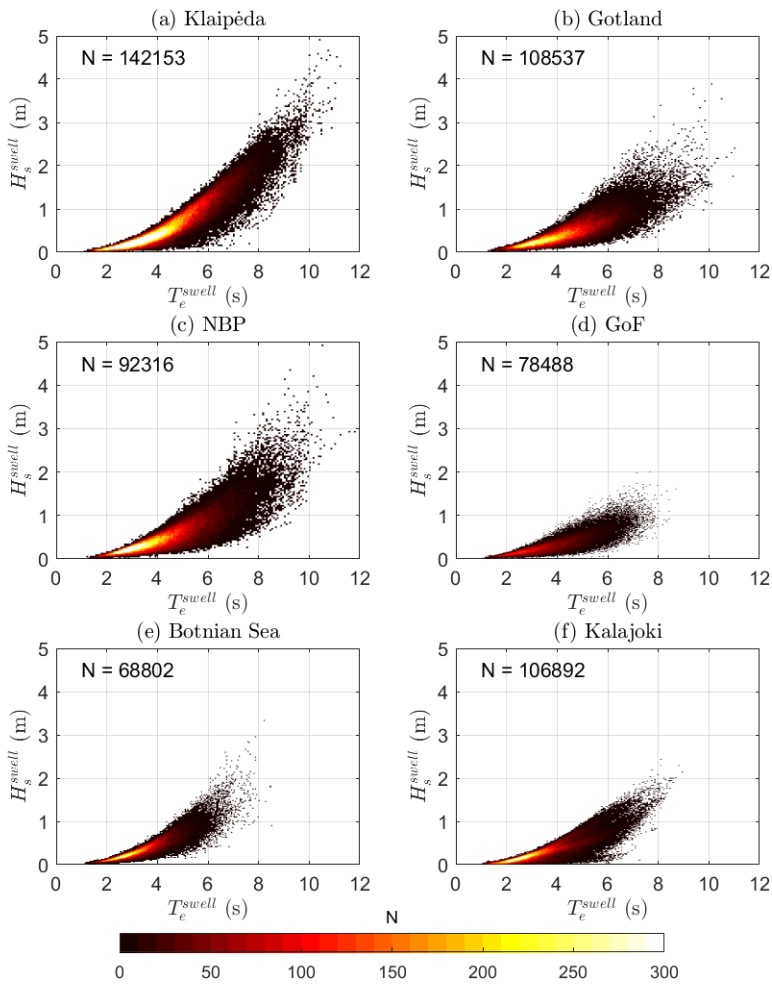

**Figure 5.** Joint occurrence of swell mean period and swell significant wave height. Cases with a swell energy weight below $W_s = 0.05$ are excluded. Ice-free statistics (Type F, Tuomi et al., 2011).

These different correlation structures offers one tool for classifying open sea and coastal locations, especially for air–sea interaction studies. The coastal locations are more heavily and constantly tainted by swell, while in the open-sea areas swell conditions are typical during the decay stage of events, leaving the growth stage free for undisturbed interaction studies.

## 4  Discussion

The energy that the wave model partitions as swell in the open sea seems to, for the most parts, be better classified as old wind-sea. Namely, locally generated wind waves turn in to swell when the wind speed decays. While these waves fulfill the

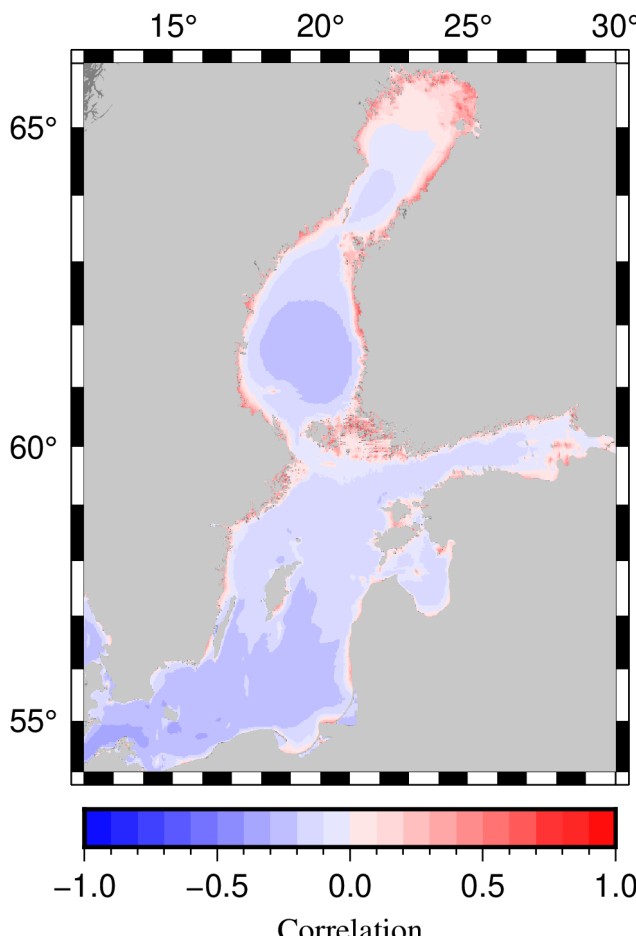

**Figure 6.** Correlation of wind-sea and swell significant wave heights. Ice-free statistics (Type F, Tuomi et al., 2011).

swell criteria, they might differ from a classical concept of swell as ordered, almost monochromatic and directional, waves generated by a distant storm. Our results show that even without remote swell the amount of swell – as defined using Eq. 5

– can still be significant (Fig. 3). Although differentiating between different types of swell is complicated, this kind of old wind-sea is bound to be present also in swell statistics compiled for the World Ocean.

In the Baltic Sea the median duration of swell events (with a height over 1 m) were under 10 h. Near the shore, however, one event at Klaipėda lasted for 86 h (between 07 December and 10 December 2017). Such long events are not residuals of high wind waves combined with a decaying wind, since the longest fetch is below 600 km. Rather, a correlation analysis (Fig. 7)

revealed that the origin of the partitioned swell energy is different near the shore. Our interpretation is that the open-sea waves – generated by the same atmospheric system – arrive to the coast already during the growth stage of the local wind sea, but are classified as swell because of the lower wind speed near the coast.

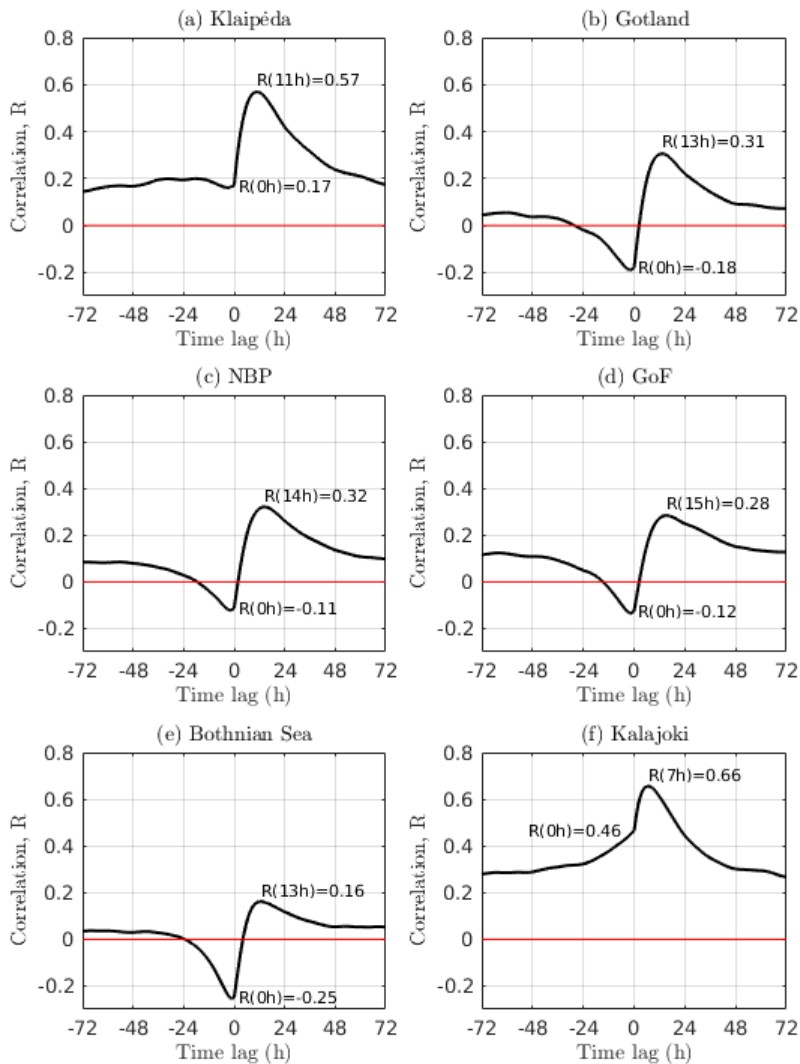

**Figure 7.** Cross-correlation functions of swell and wind-sea significant wave height calculated at six locations (see Fig. 3a). Klaipėda (a) and Kalajoki (b) are nearshore locations. A positive time lag means that swell comes after the wind sea. Ice-free statistics (Type F, Tuomi et al., 2011).

Post and Kõuts (2014) analyzed 30 cyclones over the Baltic Sea, and their average speed of 40 km h$^{-1}$ was 2–3 times faster than the deep water group speed of 5–8 s waves. For 5 s waves and a 200 km fetch the difference in arrival time would be roughly 9 h. In storms the wave period exceeds 10 s in the Baltic Sea (Soomere et al., 2008; Björkqvist et al., 2017; Björkqvist

et al., 2020), thus having a deep water group speed above $28 \text{ km h}^{-1}$. Nonetheless, the more severe cyclones that were analyzed by Post and Kõuts (2014) had average speeds of 53–59 $\text{km h}^{-1}$. The difference in arrival time (500 km fetch) between 10 s waves and a $55 \text{ km h}^{-1}$ cyclone would then also be around 9 h. These simple calculations are in the same order as the dominant 7 h and 11 h time lags found for the two coastal locations (Fig. 7), but further in-depth studies are required to further
investigate this matter quantitatively.

Our tentative classification of open sea and coastal areas through the correlation of wind-sea and swell is well motivated from the point of view of air–sea interaction studies. In such studies a detachment of the wind sea and the swell (negative correlation) is preferred, since it allows for more opportunities to study air–sea interaction processes without the results being tainted by simultaneous swell. This detachment of wind-sea and swell was identified in a study into air–sea momentum transfer
during a long, well defined, swell case at the Östergarnsholm weather station outside of Gotland (Smedman et al., 1999). The Östergarnsholm tower is an established part of the Baltic Sea research infrastructure, and our results show that the location where the Gotland wave buoy is moored is in the open sea when using this correlation-based criterion.

The sea state in larger parts of the Baltic Sea was swell dominated for about a third of the hindcast period (Fig. 3b). The winter average swell weight was below 0.3 (Fig. 3a), which is similar to the winter average for the Black Sea (Van Vledder and
Akpınar, 2016), but lower than in the North Sea (ca. 0.4-0.6, Semedo et al., 2015) and the Norwegian Sea (ca. 0.6-0.7, Semedo et al., 2015). In the World Ocean the average swell weight around the equator can exceed 0.9 during the winter (Semedo et al., 2011). The swell weights in the Baltic Sea during the summer months are slightly higher than during the winter, which is also the case for the North Sea and the Norwegian Sea (Semedo et al., 2015). The Baltic Sea swell climate differs from the World Ocean by typically being very short (under 5 s, Fig 5). In other words, our quantitative results show that high and long swell
is not persistent in the Baltic Sea (Table 1 and Fig. 3c&d), which confirms the existing qualitative understanding of the Baltic Sea wave community.

Our results differ slightly from those of Semedo et al. (2014) that were based on the 10 km resolution NORA10 product (Reistad et al., 2011). The largest differences can be seen for the swell weights near the coast, simply because NORA10 can't capture the nearshore areas. In the southeastern part of the Baltic Proper (near Kaliningrad) the results of Semedo et al.
(2014) also showed clearly higher mean swell weights compared to other part of the basin. In our results the higher values are concentrated to the coastline, and we surmise that the area of higher swell weights in NORA10 might be caused by a local underestimation of the wind speed. In the open-sea areas the winter swell weights of Semedo et al. (2014) are around 0.3 in the main basin, being in general agreement with our results.

In coastal areas swell typically drives wave-induced sediment erosion, since long swell waves reach deeper than shorter
wind waves. In our results swell was dominant roughly 70 % of time in long nearshore areas, indicating that swell plays an important role in the coastal sediment erosion also in the Baltic Sea. Still, the mean swell periods were typically short, being comparable to those of the mean wind wave periods. Also, the wind wave heights can exceed the swell wave heights near the shore. The positive correlation between wind-sea and swell waves in coastal areas poses a challenge for quantifying wave–seabed interaction, especially if waves are estimated using simple relationships of wind speed and fetch (e.g. Isæus,

2004). These considerations – together with the challenges posed by coastal archipelagos (Björkqvist et al., 2019) – means that further studies into wave–bottom interactions in the Baltic Sea are needed.

In remote sensing applications the sea state affects Synthetic Aperture Radar (SAR) back-scatter and the signal of microwave sensors (e.g., Quilfen et al., 2004; Hwang and Plant, 2010; Stopa et al., 2017). Stopa et al. (2017) linked the accuracy of the wind speed retrieved from SAR-data to both the total significant wave height and the swell height. Although the authors found that the swell height was not the main explaining factor in the accuracy of winds retrieved form scatterometers, Wang et al. (2014) showed that altimetry-based wind estimates were significantly more accurate when swell was absent. New high-resolution altimeters, such as Sentinel-3, have also brought a concern that swell can affect existing algorithms used to estimate wave heights, especially since the altimeter waveform is particularly noisy when long waves are present (Moreau et al., 2018). Furthermore, the presence of well-developed swell has been used to explain high-frequency signals in sea-level anomaly products derived using SAR-altimetry (Rieu et al., 2021). While the sea state seems to explain a significant part of the variability in several remote sensing applications, waves are unfortunately not considered in all available products, e.g. in the CMEMS scatterometers wind stress product (Driesenaar et al., 2020). The absence of dominant persistent swell might make sea-state corrections more straightforward in the Baltic Sea, especially in the open-sea areas where swell is negatively correlated to the wind sea. At the same time, globally developed algorithms might therefore need to be recalibrated for the Baltic Sea.

Classifying swell is not straightforward. We used the simplest ECMWF partitioning that only separates the wave field into wind-sea and swell based on the wave age of each wave component. More elaborate schemes that are capable of identifying several swell systems have been developed (e.g. Portilla et al., 2009), and such methods have been tested in the Black Sea by (Van Vledder and Akpınar, 2016). We found that in the Baltic Sea even ECMWF's simple partitioning flagged a minimal amount of energy as swell during the growth phase of the wave field, resulting in spurious non-zero swell weights (e.g. Fig. 2). The experimental results by Kahma et al. (2016) indicated that setting the criterion for swell as $c/U = 1$ instead of 1.2 might be more appropriate. However, these results were derived using omnidirectional spectra – thus including slower components because of the directional spread – and are therefore not directly applicable for partitioning directional spectra. Indeed, reducing the constant from 1.2 to 1 in Eq. 2 would lead to even more energy being flagged as swell during the growth stage. The coastal geometry in the Baltic Sea is challenging for spectral partitioning, and we surmise that the small artefacts of the partitioning was caused by a slight misalignment between the wave and wind direction, perhaps made worse by the 15 degree directional resolution of the spectrum. Nonetheless, this issue seems to be a harmless artefact, and the relative simple structure of the swell in the Baltic Sea open-sea areas doesn't warrant in-depth studies into more elaborate swell partitioning schemes.

One approach to classify the entire sea state (as opposed to a single wave component) as swell is to use a lower threshold value for the inverse wave age, $c_p/U$ (e.g. 1.2, Semedo et al., 2011). Nonetheless, the peak period near complex Baltic Sea shorelines might not be well defined or carry the meaning we normally attach to it (Björkqvist et al., 2019). We therefore defined the sea state to be swell dominated if over half of the energy was classified as swell ($W_S$>0.5); this metric is well defined even in complex coastal conditions, although the 1 nmi wave product used in this paper was too coarse to reliably study waves in these types of areas. Our probabilities for swell dominated sea states (Fig. 3c&d) mostly coincide with those of Semedo et al. (2014) (calculated using the $c_p/U$ definition), meaning that these two definitions have a general agreement,

at least in the mean sense. The inverse wave age (in its most simplest form) doesn't account for directionality, but the effect of swell–wind angles is not expected to be a dominant process in e.g. upper layer mixing in the Baltic Sea because of the absence of a persistent independent swell component. Rather, these applications might be more heavily influenced by the persistent misalignment of wind and waves caused by slanting fetch (Pettersson et al., 2010).

The seasonal ice-cover means that that there is no single way to calculate wave statistics in the Baltic Sea (Tuomi et al., 2011). However, since the swell weight, $W_S$, is not defined if $H_s = 0$, only ice-free statistics (Type F) could be compiled for this parameter. We also note that the inverse wave age, $c_p/U$, and even the peak period are ill-defined for the ice-time. We chose to present the swell height results using ice-included statistics, which are better suitable for e.g. wave energy studies or ocean engineering fatigue calculations. The results show that swell in the Baltic Sea is intermittent, which means that it cannot provide a continuous source of energy even in areas where the sea is typically not ice-covered. Type F statistics would have been more relevant for air–sea interaction studies, but the difference between these two statistics are almost non-existent below roughly 60° latitude (Björkqvist et al., 2018). Our results for the Baltic Proper are therefore not affected by the seasonal ice-cover to any meaningful extent.

## 5 Conclusions

We studied swell in the semi-enclosed Baltic Sea using 20 years of wave model data. Our data originated from the same wave simulation that was used to compile the CMEMS wave product BALTICSEA_REANALYSIS_WAV_003_015, but also included partitioned total swell parameters. The partitioning scheme used a typical criteria based on the wave phase speed relative to the wind speed (see Eq. 2). While this definition has a physical motivation, it also identifies waves that do not fit our conceptual, somewhat idealised, view of long crested, monochromatic and unidirectional wave trains. Nevertheless, we determined the swell energy weight (the fraction of wave energy the model considered swell) following Semedo et al. (2011), and classified the sea state as swell dominated if over half the energy was swell.

Swell in the Baltic Sea was typically low (under 2 m) and short (under 5 s), with the mean swell significant wave height in the open-sea areas being below 0.4 m both in the winter and summer seasons. The probability of the sea state in the open sea being swell dominated was below 30 % (winter) and 50 % (summer); the average swell weight was below 0.3 during the winter, and mostly under 0.4 in the summer. Both swell height and swell prevalence for the open-sea areas in the Baltic Sea was therefore significantly lower than for the World Ocean.

Areas with high swell prevalence was found near the coast in every sub-basin except the Gulf of Riga. Swell was highest and most prevalent in the southeastern part of the Baltic Proper, at the Lithuanian coast outside of Klaipėda and the coast off Kaliningrad. The swell dominated cases were still mostly caused by a low wind speed ($U > 5 \text{ m s}^{-1}$) rather than especially high swell heights.

The Baltic Sea can be classified into open sea and coastal areas if the overall correlation (positive–negative) is used as a proxy for the two different cross-correlation structures between wind-sea and swell wave heights. In the open sea the correlation was negative, because swell was mostly created simply by a decaying wind, turning the existing wind-sea waves to swell in the

partitioning. Swell and wind sea heights had the strongest positive correlation for a 15 h time lag, which roughly corresponds to the growth-and-decay cycle of waves generated by a weather system. For the coastal locations the correlation was positive, meaning that swell energy was not simply created from old wind sea. The cross-correlation was strongest for a ca 10 h time lag, which we interpret as the difference in arrival time between a weather system that generates local waves and the arrival of longer waves generated by that system in the open sea; these longer waves are classified as swell near to coast because the wind weakens towards the coast.

Air–sea interaction studies should be made in areas where the correlation is negative. The positive correlation near the coastlines is a sign of more complex dependence between swell and wind-sea waves; this problem will be relevant when studying wave–seabed interactions.

*Data availability.* The data needed to reproduce the figures and the tables will be made available as a supplement to the article in upon publication of the final manuscript.

*Author contributions.* The study was initiated by JVB and VA, and the conceptualization was developed further by JVB, VA, and SR. The review of the literature and the visualisation was performed by SP and JVB. The model data was produced, maintained and processed by EL and LT. The analysis of the data was performed by JVB, with input from SP and VA. The manuscript was prepared by JVB with contributions from all co-authors.

*Competing interests.* The authors declare that no competing interests are present.

*Acknowledgements.* This work was partially supported by the Personal Research Funding of the Estonian Ministry of Education and Research (grant number PSG22) and the Copernicus Marine Environment Monitoring Service (CMEMS). Hersbach et al. (2020) was downloaded from the Copernicus Climate Change Service (C3S) Climate Data Store. The results contain modified Copernicus Climate Change Service information 2019. Neither the European Commission nor ECMWF is responsible for any use that may be made of the Copernicus information or data it contains. We thank the reviewers for their comments and suggestions that helped us sharpen the message of the article.

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
