# Peer review of "Swell hindcast statistics for the Baltic Sea"

_Ocean Science, 2021_

## Author Comment (AC1)

**Response to Referee #1**

Jan-Victor Björkqvist et al., 24.9.2021

The referee comments are marked in blue italics, while our responses are in normal font.

*General comments*

*R#1: I have twofold feelings about this manuscript.*

*On the one hand, it is a nice insight into the structure of wind waves in the Baltic Sea that highlights several interesting features, such as short duration and thus great intermittency of formal swell events, extensive spatial variation in the probability of predominance of formal swell events, low swell heights, overall short periods of swell waves that almost overlap with the typical periods of windseas (most frequently 2–4 s), frequent co-presence of windsea and swell.*

*On the other hand, some established features strongly signal that attempts of partitioning of the Baltic Sea wave fields into windsea and swell do not necessarily lead to sensible results and may even be deceptive. Even though many such aspects are discussed professionally, the ramifications of possible misunderstandings are not really made clear for the reader.*

**Our response:** Thank you for agreeing to review our manuscript. It is greatly appreciated. We agree that swell in the Baltic Sea might not fulfill the expectations one might have from looking at swell in oceanic conditions. We still have to disagree with the statement that the results would not be sensible. That said, the distinction you raised in the comments should be better clarified to avoid confusion. We have reworked the manuscript to this end. Please also see our response to the first specific comment.

*Specific comments*

*R#1: One of the main results is that the areas where swell wave energy predominate form only a narrow strip along some 50% of the length of coastlines of the Baltic Sea (Fig. 3). This feature is counter-intuitive because energy of (longer) swell waves should decay faster in the shallow area than energy of (shorter) windseas. Also, the (longer) swell waves experience stronger refraction in areas with variable depth than (shorter) windseas. This effect also leads to systematically faster decrease in the energy of swell waves in the nearshore. Differently from the open ocean conditions, the Baltic Sea waves often approach the shore under relatively large angles. It is thus likely that this effect is stronger in the Baltic Sea compared to its impact on the open ocean shores.*

*As I am sure that there is nothing wrong with the simulations and the evaluation of the share of swell, the results essentially prove that the definition of swell (that has been derived for the open ocean conditions) is simply not applicable for the Baltic Sea. This*

*aspect should definitely be clarified for the benefit of readers outside the Baltic Sea basin.*

*An attempt to explain this feature in the last paragraph of Section 3.4 (lines 164–173) is somewhat unfortunate and basically expresses the same point. In particular, I strongly disagree with the statement on line 174 that "The coastal locations are more heavily and constantly tainted by swell" as practically the same wave system that exists a dozen km from the shore approaches the coast – and is just renamed swell because a decrease in the wind speed in the nearshore. This aspect is mildly expressed in Discussion on lines 182–184 but not really made clear.*

*A possible misunderstanding is reinforced in Conclusions by saying on line 283: "swell was mostly created simply by a decaying wind, turning the existing wind-sea waves to swell in the partitioning". It would be correct to say: the wind-sea was just interpreted as swell in the nearshore in the used framework even if its properties did not change. Also the conjecture that the probability of having swell-dominated wave fields (expressed in terms of the correlation between swell and windsea height) substantially changes from offshore to nearshore belongs to the same pool of ideas and apparently reflects the "impact" of the particular swell definition.*

*In this context, part of Discussion is also deceptive. The text on lines 213–218 relies on the usually distinctly different properties of swell and windseas in the open ocean. The presentation has made clear that "swell waves" is just a name for the same wave system, with the same period; just wind has ceased. Also, Wang et al. (2014) look at the global ocean where swell waves are usually longer and well organized, closed to monochromatic ones – that is typically not the case of Baltic Sea swell (as proved by the authors). The real point to address is formulated on lines 236–237: "We found that in the Baltic Sea even ECMWF's simple partitioning flagged a minimal amount of energy as swell during the growth phase of the wave field, resulting in spurious non-zero swell weights".*

*In general, it seems that the authors are partially victims of the major success of high-resolution wave simulations in the Baltic Sea. The spatial resolution is now so high that effects that are not visible in global wave hindcasts start to play a large role in the interpretation of the local results. The overarching conjecture from the manuscript could perhaps formulated as follows: the existing separation methods of windsea and swell make only sense for offshore conditions, and should not be used in the nearshore and in archipelago areas.*

*Based on the above considerations, I recommend reshaping the discussion and conclusions so that the situation would be unambiguously clear for non-experts in the field. It may make sense to add a few sentences that clarify the difference between citizens' perception of swell as organised, highly directional, and often almost monochromatic wave field and the wave modellers' understanding of swell as some collection of wave components.*

**Our response:** We agree that the difference between the conceptual perception of swell waves doesn't necessarily coincide with the quantitative definition used in e.g. wave models. However, the quantitative definition still has real physical meaning, and the conceptual definition is hard to quantify. In addition, the results derived from the World Ocean will also include these "old wind sea" events, even though a large part of the swell energy might come from waves that also fulfill the more subjective definition.

We do not agree that the definition of swell is "not applicable" in the Baltic Sea, since the definition is based on a real physical criteria that affects e.g. wind-wave growth. Our results simply show that the types of waves that fulfil the generally accepted definition of swell are not similar to the typical conditions in the oceans. We don't agree that the minimal energy that the ECMWF criteria flags as swell is a serious issue, since it has no significant impact on the results. Also, these artefact might well exist also elsewhere, but because of its minimal amount it is revealed only in sheltered areas that can be completely free from remote swell. It more shows that the quantification of swell is a complicated issue (e.g. Portilla et al., 2009), and that using even more elaborate schemes to separate swell is probably not warranted in the Baltic Sea.

The question of if waves are simply "renamed swell" after the wind ceased, or if they "become swell" after the wind ceased is almost a semantic argument. As mentioned, we do still agree with your point that this distinction should be made, and especially the last sentence of your comment nailed it. We have revised the manuscript, starting from the abstract and introduction, to 1) make clear that this distinction between "conceptual swell" and "quantitative swell" exists, and 2) make clear that we are using the latter definition in this paper. We have also modified the Discussion and Conclusions to reflect this. The main additions to the manuscript are:

Start of the **abstract**:

*"The classic characterisation of swell as regular, almost monochromatic, wave trains doesn't necessarily accurately describe swell in water bodies shielded from the oceanic wave climate. In such enclosed areas the locally generated swell waves still contribute to processes at the air and seabed interfaces, and their presence can be quantified by partitioning wave components based on their speed relative to the wind."*

Added paragraph to the **introduction**:

*"Conceptually swell is often thought of as regular and long-crested waves that are almost monochromatic and highly directional (e.g. Holthuijsen, 2007, page 47). In the World Ocean this characterization is apt (Barber and Ursell, 1948), and might also coincide with a layperson's view on swell. Nonetheless, in wave measurements and wave models swell needs to be quantified, and the definition of swell is typically (loosely speaking) taken as waves outrunning the wind (e.g. Bidlot, 2001). As a result, swell is taken simply as a selection of wave components that fulfil this criteria. While the conceptual regular swell waves also fulfil the quantitative criteria, the opposite is not necessarily true – although this is subjective. Be that as it may, the quantitative criteria is*

*still well motivated from a standpoint of air–sea interaction, including wind-wave growth (e.g. Komen et al., 1984; Kahma et al., 2016)."*

Addition to first paragraph in **discussion**:

*"While these waves fulfill the swell criteria, they might differ from a classical concept of swell as ordered, almost monochromatic and directional, waves generated by a distant storm. Our results show that even without remote swell the amount of swell – as defined using Eq. 5 – can still be significant (Fig. 3). Although differentiating between different types of swell is complicated, this kind of old wind-sea is bound to be present also in swell statistics compiled for the World Ocean."*

Addition to the first paragraph of the **conclusions**:

*"The partitioning scheme used a typical criteria based on the wave phase speed relative to the wind speed (see Eq. 2). While this definition has a physical motivation, it also identifies waves that do not fit our conceptual, somewhat idealised, view of long crested, monochromatic and unidirectional wave trains. "*

*R#1: Technical issues: A short (not full) list of minor aspects that might need attention:*

*Line 38: The reference to Semedo et al. (2014) is to a certain extent deceptive as this source is visible only as an abstract. Also, the poster available in internet has a different team of authors: Alvaro Semedo, Roberto Vettor, Oyvind Breivik, Andreas Sterl, Magnar Reistad, Daniela C.A. Lima, Carlos Guedes Soares.*

**Our response:** Thank you for pointing this out. Since all of the authors listed in the abstract are also listed on the poster (but not vice versa), we decided to remove the details concerning the abstract. The citation is therefore now unambiguous and all authors are still credited for the work. We acknowledge that posters are typically not cited, but in this instance there exists no proper literature that could be cited instead.

*R#1: Lines 45–46: The information on these lines is highly cryptic. Please explain the acronyms.*

**Our response:** We have now explained the acronyms.

*R#1: Line 51: The acronym ERA5 also requires an explanation and a reference.*

**Our response:** We have added the explanation and the reference to the article.

*R#1: Line 84, Equation (5) and also below: it is recommended to unify the use of "S"/"s" as subscript for quantities that characterize swell.*

**Our response:** The only variable that uses a subscript to characterize swell is the swell weight, WS. The use of a capital S here is in line with Semedo et al. (2011). In other cases we explicitly spell out "swell" or "sea" as a superscript, with a lowercase "s" simply signifying that it is the significant wave height. The use of superscripts is also in line with

Semedo et al. (2011), although they use "s" and "w" instead of "swell" and "sea". Nonetheless, we see no possibility for confusion here, and have therefore decided to keep the notation as is.

*R#1: Line 85: probably "dominated".*

**Our response:** Yes, this was indeed what we meant. It has now been corrected to the manuscript.

*R#1: Line 99: it is recommended to use the Lithuanian "dotted e" in KlaipÄ–da.*

**Our response:** Thank you for bringing this to our attention. This has been corrected throughout the manuscript, including the figures.

*R#1: Line 107, also caption to Fig. 1, line 111, 114: the superscript "th" should use normal font, not italics.*

**Our response:** We found five instances of this in the text. They have all been corrected.

*R#1: Figure 1, explanations to both scales: "H_s" should be in italics; also the upper scale should have a space between H_s and (m).*

**Our response:** This has been corrected in the figure.

*R#1: Caption to Fig. 2: should be "storm"*

**Our response:** Thank you for pointing this out. We have now correct *"took plave during the sotrm"* to *"took place during the storm"*.

*R#1: It is recommended to use "Baltic proper" as it is not really a proper name; however, there exists also a tradition of capitalizing "Proper".*

**Our response:** Our group has had the convention of using the capitalized form, and we therefore prefer to keep it as it is. Although we do appreciate you bringing this to our attention.

*R#1: Line 118–119: "the averaged swell direction were" is inconsistent*

**Our response:** This has now been changed to "averaged swell directions were"

*R#1: Lines 128–129: the conjecture "In the Baltic Proper the highest swell weights were along the eastern coastlines, which is expected because of the prevailing southwesterly winds (Karagali et al., 2014)" is only partially true. In fact, the NNW winds also add substantially to the wave fields so that the overall maximum wave height (Björkqvist et al., 2018) and particularly the maximum of wave energy flux (Nilsson, E., Rutgersson, A., Dingwell, A., Björkqvist, J.V., Pettersson, H., Axell, L., Nyberg, J., Stromstedt, E. 2019. Characterization of wave energy potential for the Baltic Sea with focus on the Swedish Exclusive Economic Zone. Energies, 12(5), 793, doi: 10.3390/en12050793) are located between Gotland and the Gulf of Gdansk. It might be also mentioned that refraction*

*plays usually a larger role in the propagation of (longer) swell than for (shorter) windseas in the relatively shallow Baltic Sea.*

**Our response:** It is true that a prevailing wind direction in itself does not necessarily mean that the highest winds (and waves) are from the prevailing direction. There is, however, a well established east-west asymmetry in the Baltic Sea wave climate, with larger waves being found more to the eastern part of the main basin (Tuomi et al. 2011, Fig. 11; Björkqvist et al. 2018, Fig. 5; and our Fig. 1c in the manuscript). This is a result of the strong winds from the sector between west to south. For the wave energy potential Nilsson et al. 2019 (Fig. 8) showed that the highest wave energy fluxes were also modelled for southerly and westerly wave directions.

It is true that the highest waves in the Baltic Sea are probably between Gotland and the Gulf of Gdansk. Nonetheless, they are typically from the west-to-south sector, thus ending up on the eastern coasts of the Baltic Sea main basin.

We have edited the sentence slightly to make it clearer, and it now reads:

*"In the Baltic Proper the highest swell weights were along the eastern coastlines, which is expected since the prevailing southwesterly winds (Karagali et al., 2014) lead to an east–west asymmetry in the wave climate of the Baltic Proper (e.g. Tuomi et al., 2011; Björkqvist et al., 2018, and our Fig. 1c)."*

We have added a mention of the stronger refraction of the swell compared to wind waves to the section dealing with wave directions.

*R#1: Line 139: please explain the abbreviations NBP and GoF.*

**Our response:** This has been corrected in the manuscript.

*R#1: Line 237: remove one "the" in the middle of the line.*

**Our response:** This has been corrected in the manuscript.

---

## Author Comment (AC2)

**Response to Referee #2**

Jan-Victor Björkqvist et al., 24.9.2021

The referee comments are marked in blue italics, while our responses are in normal font.

*R#2: Swell waves play a critical role in air-sea interactions. In this study, some interesting results of the swell in the Baltic Sea are drawn based on 20 years of high-resolution wave simulation data. These results are interesting to the Baltic Sea research community.*

**Our response:** Thank you for taking the time to review our manuscript. It is much appreciated.

*I have the following comments/suggestions about the study.*

*General comments:*

*R#2: The wind condition varies significantly with the season which may result in the variation of swell probability and energy weight. Authors give the swell height distribution in winter and summer. I am wondering if there are any seasonal variations of the swell energy weight, swell probability, swell period. If so, I would recommend including those analyses?*

**Our response:** Thank you for this comment. We have updated the swell prevalence figure to show the seasonal variation of the swell energy weight and probability of swell dominance. Including the seasonal variation also for these variables makes our results more comparable to those of Semedo et al. (2011, 2015) for the World Ocean and Nordic seas. The updated figure is shown as Fig. 1 in this document, and will also be updated to the revised manuscript. The results, discussion and conclusions have been rewritten to incorporate the new seasonal results.

The relation between the periods and the significant wave height did not show any interesting seasonal variation. Longer waves are generated (and thus turned to swell) during the winter compared to the summer, but this fact is hardly surprising. The seasonal figures are shown in Fig. 2 & 3 of this document, but the figure in the manuscript was kept as is, since we feel that the seasonal variations did not warrant splitting the scatter plots up to several figures.

[Figure]

*Fig. 1. The updated Fig. 3 of the manuscript where the swell weight and probabilities are separated by season.*

[Figure]

Fig. 2. Same as Fig 4. in the manuscript but only for the winter season (DJF).

[Figure]

*Fig. 3. Same as Fig. 4 in the manuscript, but only for the summer season (JJA).*

**R#2:** *In the introduction, the authors point out the importance of the misalignment between wind and swell direction (L20). With the data, I think the authors can analyze the distribution of the swell-wind angle. If so, I would suggest the author add one section about it.*

**Our response:** We calculated the distributions of swell-wind directional misalignments for the six locations. For all data (including also very low swell heights) the general distribution of misalignment is the same for almost all locations (Fig. 4 in this document). For the 99th percentile of the swell height we can see that the open sea points in the main basin behave differently from the coastal points. Namely, the most probable

amount of misalignment in the open sea is 60-100 degrees, while at coastal locations the most probable misalignment is below 20 degrees.

These results reinforce the correlation analysis that the open sea and coastal zones have different generation mechanisms for swell. In the open sea high swell waves are generated when a weather system passes and the wind turns (and the wind speed simultaneously drops), as seen in e.g. Fig. 2 of the manuscript. In the nearshore areas the wind can simply attenuate near the coast, causing an increase in the wave energy classified as swell even though the weather system has not passed, and the direction might be well aligned with the wind direction.

We have added a separate section for the wind direction, and included the below figure and the accompanying analysis there.

*R#2: In section 3.4, the authors give some interesting results about the correlation of wind-sea and swell. The negative correction is contributed to the decaying wind. Based on Eq. 2, the wind direction is also an important factor determining if a wave mode is swell or wind wave. Did the authors have some analysis about the contribution of the variation of wind direction? If you look at Fig 2, the wind direction change is also significant for the variation of swell and wind wave height.*

**Our response:** Because of the cos-term in Eq. 2, the relative wind-wave direction is definitely an important factor. Typically drastic and sudden changes in the wind direction is expected to be related to the passing of a weather system. This is expected to be seen simultaneously also in the wind speed (see e.g. Fig. 2 in the manuscript). The analysis in the swell-wind misalignment (see Fig. 4 of this document) suggests that for the highest swell heights the directional difference between the swell waves and wind are large in the open sea. At the coast, again, the directional difference is typically small.

However, just because the directional difference is large, that doesn't mean that it would be the dominating factor. Even for a 90 degree difference the wind speed might simultaneously be low enough that a significant amount of energy would be classified as swell even if the wind and swell waves were perfectly aligned. In other words: what is the main reason for swell is the directional difference is close to 90 degrees and the wind speed is close to 0? Differentiating between the contributions of these two correlated sources is tricky. We therefore decided not to pursue this type of analysis beyond that presented in Fig. 4 and the newly added section on swell direction. The main results show the combined effect of these two connected phenomena (wind speed and direction) through the wind speed being projected to the wave direction in Eq. 2.

[Figure]

Fig. 4. The probability of differences in swell-wind angles. Swell weights over 0.05.The red bars signify cases where the swell height is above the 99th percentile.

*R#2:* *In the analysis, the correlation is used to show the relation between swell and wind. For the correlation section, why use wind-sea wave height, but not wind speed? They may do not show a significant difference since wind wave height has a highly linear relationship with the wind.*

**Our response:** The correlation analysis was not done to show a relation between the wind speed and the swell, but rather the wind-sea and the swell. The idea is that in oceanic conditions (for distant swell), the swell height would be uncorrelated with the local wind sea. However, since distant oceanic swell is absent in the Baltic Sea, the question arose to which extent the swell is "distant enough" to be detached from the locally generated waves. This analysis gave us some insight into the structure of the wave field in the Baltic Sea, as presented in the results and discussion.

As you suspected, the strong connection between the wind speed and the wind-sea height means that the correlation structure calculated between the wind speed and the swell height is very similar to the results presented between the wind sea and swell (see Fig. 5 of this document). Since we were, in the end, interested in the structure of the wave field, we feel that including the wind speed at this stage obscures the point, although it would ultimately lead us to the same conclusions.

[Figure]

*Fig. 5 The correlation between the swell height and the wind speed. The results are very similar to the correlations between the wind-sea height and the swell height.*

***R#2:*** *Minor comments:*

- *L10: suggests→suggest*

**Our response:** This has been corrected.

- *L19: upwards→upward*

  **Our response:** This has been corrected.

- *L55: What is the temporal resolution of the ice data?*

  **Our response:** The temporal resolution of the ice data varies. Between 1992 and 2005 the resolution is 3-4 days (updated twice a week) and from March 2005 onward the resolution is mostly 1 day. This information has been added to the manuscript.

- *Figure 2: Give the full words of the abbreviations NBP*

  **Our response:** This has been corrected in the figure caption.

- *Figure 4: Authors can consider using the normalized distribution since the total data points vary with stations which may be easier to show the distribution (in particular for GoF). The total number of data points should be given.*

  **Our response:** In the end we chose not to use a normalized distribution, since the current figure also contains visual information about the differences in the swell climate between the locations, which would be lost in the normalization. The total number of points have been added to the plot.

---

## Author Response (AR2)

**Response to review comments from Referee #1 to the manuscript "Swell hindcast statistics for the Baltic Sea" (OS-2021-62)**

Jan-Victor Björkqvist et al., 25.10.2021

The referee comments are marked in blue italics, while our responses are in normal font.

*I appreciate the efforts of the authors to improve the manuscript and to adequately respond to the comments and suggestions by all three referees.*
*There are still a couple of issues that need attention plus a few mostly technical aspects and typos.*

**Our response:** Thank you for following up and reviewing the updated version of the manuscript. Please find our comments to the suggestions below.

*Issues that may affect the content:*

*Line 138: It would be definitely correct to say that the largest swell HEIGHTS were expected at the eastern coastlines; however, the outcome that the highest swell WEIGHTS are also there is quite unexpected – at least to my eyes. This result to a large extent relies on the difference of translation speed of cyclones in this area. This is one of the main conclusions of the work and is explained much later in the manuscript.*

**Our response:** You are right that we probably drew too quick conclusions here. The stated fact does require some additional explanation. Since this is in the results section, and the entire mention is not central, breaking up the flow of the text with a detailed analysis is probably not a good trade off. We have therefore simply dropped this claim altogether, with the two sentences now reading:

*"In the Baltic Proper the highest swell weights were along the eastern coastlines. Nonetheless, also a short coastal section in southeastern Gotland had mean swell weights exceeding 0.6."*

This now connects the two relevant pieces of information in two following sentences instead of separating them with a long "parenthetical" statement.

*Line 173–174: To my eyes, the use of "erratic statistics" is misleading here. The image shows mismatch between the two data sets and a two-peak structure of one of these. Also, probably it is meant that the cases shown on the image are from two (or more) different populations with possibly different distributions. Finally, please adjust "a results".*

**Our response:** Thank you for pointing this out. We have reformulated this sentence and it now reads

*"The distribution at Kalajoki (f) in the Bay of Bothnia forms an exception, which we surmise is because the ice-cover changes the fetch geometry; the data are thus essentially not from a single population."*

We have removed the mention that it is near a sandy beach, since it is not relevant. The relevant part is that it is located in the Bay of Bothnia (because of the ice-cover).

*Lines 192–193: the sentence "The negative correlation of the open-sea areas were indicative of decaying winds turning existing wind-sea into swell in a zero-sum fashion" is cryptic, grammatically incorrect ("were") and conceptually inexact. The negative correlation may also stem from rapid changes in the wind direction (see Eq. (2)). In other words, it may be associated (at least to some extent) with small size of high-latitude cyclones. The windsea doesn't turn into swell, it is just interpreted as swell based on Eq. (2). It is done so that the total energy of the wave system remains constant. "Zero-sum" is simply wrong in this context as only the partitioning changes.*

**Our response:** We have added the mention that it might also be caused by turning winds (i.e. decaying projected wind speeds). The new sentence (which also addresses the other parts of your comment) now reads:

*"The negative correlation of the open-sea areas was indicative of decaying or turning winds that caused existing wind-sea to be reclassified as swell."*

*Lines 254 and 256: It is incorrect to only mention erosion as swell waves are one of the major drivers of beach recovery after strong impact of windseas. Please use: entrainment, transport, motion, or even all three aspects.*

**Our response:** We have changed this to the more general *"sediment transport".*

*Line 257: As there seems to be no large difference between swell and windsea properties, I strongly recommend adding an explanation, such as "Therefore, differently from the situation in the World Ocean, the partitioning [of wave field into swell and windseas] has limited physical significance in the context of coastal processes."*

**Our response:** Thank you for this comment that gave us the chance to sharpen this part of the manuscript. We have reformulated this part of the manuscript and it now reads:

*"In our results swell was dominant roughly 70 % of time in long nearshore areas. The mean swell periods were typically short, being comparable to those of the mean wind wave periods, and the wind wave heights can exceed the swell wave heights near the shore. Therefore, the physical significance of swell partitioning for coastal processes in the Baltic Sea is limited compared to the World Ocean. Nonetheless, the positive correlation between wind-sea and swell waves in coastal areas poses a challenge for quantifying wave–seabed interaction, especially if waves are estimated using simple relationships of wind speed and fetch (e.g. Isæus, 2004). These considerations – together with the challenges posed by coastal archipelagos (Björkqvist et al., 2019) – means that further studies into wave–bottom interactions in the Baltic Sea are needed."*

*Line 279: the word "spurious" is incorrect. Windseas rapidly fill with some energy the entire directional spectrum of wave components, and some of them fall inevitably into the category of swell owing to Eq. (2). Therefore, after a few steps of running WAM with any input*

*nonzero wind field you will see non-zero swell height at every grid point. This is an intrinsic property of Hasselmann's equation.*

**Our response:** We disagree on this point. In the numerical wave models a large part of the spectra can be (practically) 0. This is in part because of the way the wind-input source term is defined (being quasi-linear with respect to the spectral energy), and probably because the weakly non-linear four-wave interactions are calculated using an approximation of discrete interactions (DIA).

In Figure 1 below we show an example spectrum. The spectrum is unrelated to the wave modelling of this paper, but illustrates this very general point. The spectrum is taken outside of the Norwegian coast from a 3 km WAM wave model run (the data has been transformed into the format for WAVEWATCH III, hence the file and variable names). This spectrum was chosen because it was readily available in binary format.

The figure clearly shows that the entire spectrum is not filled with energy, since there is a remarkable difference between the max values (over 40) and the minimum values (order $10^{-16}$). The logarithmic color scale on the right also illustrates that the very low spectral density is not an isolated point, but covers a large part of the spectrum. There therefore exists no "background" values for the entire spectrum that would result in quantifiable swell height in a general sense. The swell heights that are seen e.g. in Figure 2 of the manuscript around 22 Dec 18:00 are caused by something other than a type of energy filling the entire spectrum that was described in your comment. This is also evident from Figure 2 in another way, since there is a visible increase in the swell heights in the middle of the growth stage: a significant amount of background energy to cause measurable swell weights would be present constantly.

The small, but measurable, swell heights that we sometimes observed in the growth stage were not caused by a constant background noise. Rather, they sometimes appear in an unexpected fashion because the swell partitioning algorithm flags a small amount of the non-zero part of the spectrum as swell, even though no swell is reasonably expected in these kinds of growth situations. This can, for all reasonable intents and purposes, be denoted as "spurious", and we therefore stand by our choice of word in this context.

[Figure]

**Fig. 1** *An example spectrum that shows that the spectral density can be zero in large parts of the spectrum. Left shows the spectrum in a linear color scale, while right shows the same spectrum in a logarithmic color scale.*

*Technical details*
*The last line of Abstract: I recommend to explicitly incorporate the point that in many occasions the wave system that is considered as swell arrives with a delay after the storm has already moved further.*

**Our response:** This seems like a good idea. We have added the following sentence to the end of the abstract:

*"Namely, the highest swell typically arrives with a roughly 10 hour delay after the low-pressure system has already passed."*

*Line 135: better say "low values" or "low estimates" rather than "low numbers".*

**Our response:** This has now been corrected to "low values"

*Line 145: It is recommended to mention that swell heights are much lower in summer than in winter; just to make clear that the formal predominance of swell-dominated conditions does not mean more swell energy in summer than in winter.*

**Our response:** We have added a mention of this at the end of the paragraph:

*"We note that the higher swell weights during the summer are not indicative of higher absolute swell heights, as evident from Sect. 3.1 (Fig. 1)"*

*Line 165: use "Bay of Gdańsk", "Gdańsk Bay" or "Gulf of Gdańsk" (all three versions are in circulation) but capitalize "Bay/Gulf".*

**Our response:** Thank you for bringing this to our attention. This has now been corrected to "Bay of Gdańsk".

*Line 167: it is recommended to mention that [longer] swell waves are usually refracted more intensively in shallow coastal areas than [shorter] windseas. The use of simply "refracted" carries no information.*

**Our response:** Indeed. We have edited this part of the manuscript and it now reads:

*"Typical swell periods in the Baltic Sea are short (see Sec. 3.4), but they are still, on average, longer than the wind-sea. The relatively shallow coastal areas therefore refract swell waves more strongly compared to wind-sea waves."*

*Lines 168–169: The use of "divergence area" could lead to misinterpretation as divergence has clear physical meaning. In essence, there is no clearly defined average swell direction in this area.*

**Our response:** Thank you for pointing this out. This has been modified to:

*"In the Bothnian Sea the averaged swell directions were towards the coast, with the averaged direction not being clearly defined in the middle of the basin - a pattern already identified by Semedo et al. (2014)."*

*Line 171: please adjust "misalignment … are".*

**Our response:** This has been corrected to:

*"The misalignment between the swell direction and the wind direction is roughly similar for all the six locations [...]"*

*Table 1: empty dimensions (-) of the dimensionless W_S could be removed.*

**Our response:** We note that the similar notation is used in Figure 3. We prefer to explicitly state the empty dimensions. Nonetheless, we didn't find any stated preference in the journal's submissions guidelines for mathematical notation and terminology. We therefore keep this as is for now, but are ready to change the notation based on the recommendation of the editor or production staff.

*Line 176: adjust "follow".*

**Our response:** This has been corrected to *"follows".*

*Line 202: wind direction may also turn near the shore (as wind systems usually have a tendency to be shore-aligned), and the use of Eq. (2) results a larger share of swell.*

**Our response:** This is indeed true. We now mention this possibility and the sentence reads:

*"These longer waves are generated by the same weather system, but because the wind decays or turns closer to the coast, they are classified as swell near the shoreline."*

*Line 207: adjust "structures offers".*

**Our response:** This has now been corrected to *"structures offer".*

*Line 208: The sentence "The coastal locations are more heavily and constantly tainted by swell …" may create a feeling that swell is very heavy in some nearshore locations of the Baltic Sea. Even though, technically, the claim is correct, I suggest to reformulate the sentence to avoid misinterpretation.*

**Our response:** This has been reformulated to:

*"The coastal locations have a more persistent swell, while in the open-sea areas swell conditions are typical during the decay stage of events, leaving the growth stage free for undisturbed interaction studies."*

*Line 212: add "or wind direction turns"; see comment to line 202.*

**Our response:** We have now corrected this sentence to:

*"Namely, locally generated wind waves turn in to swell when the wind speed decays or the wind turns."*

*Line 217: adjust "duration … were".*

**Our response:** This has now been corrected to *"duration […] was"*

*Line 220: it is correct to say "the origin"; however, the wording might additionally include an expression like "alternatively, the reason why" or similar.*

**Our response:** We are slightly unsure about what is exactly meant by this comment. Since the reason for the different origin is already elaborated on in the following sentence, we decided to keep the wording here as it is.

*Line 223: it is recommended to speak about "relocation speed" or similar, to make clear difference from wind speed.*

**Our response:** We have changed this to *"translation speed".*

*Line 227: see comment to Line 223.*

**Our response:** This has also been corrected to *"translation speed"*.

*Line 228 The expression "These simple calculations are in the same order …" does not make sense. Please consider saying "These simple calculations suggest the delay time that is …" or similar.*

**Our response:** Thank you for pointing this out. The sentence has now been corrected to:

*"The delay times estimated by these simple calculations are thus in the same order as the dominant 7 h and 11 h time lags found for the two coastal locations[...]"*

*Lines 242–243: similarly to the comment to Line 145: it is recommended to mention that swell heights are much lower in summer than in winter; just to make clear that the formal predominance of swell-dominated conditions does not mean more swell energy in summer than in winter.*

**Our response:** This has been modified to:

*"The swell weights in the Baltic Sea during the summer months are slightly higher than during the winter (vice versa is true for the actual swell heights), which is also the case for the North Sea and the Norwegian Sea (Semedo et al. 2015)."*

*Line 250: adjust "part" (parts?).*

**Our response:** This has been corrected to *"parts"*

*General remark: both "e.g." and "e.g.," are used.*

**Our response:** We have edited the manuscript to use *"e.g."* consistently.

*Line 276: please add that Eq. (2) also takes into account the difference in wind and wave directions.*

**Our response:** This has been amended to:

*"Classifying swell is not straightforward. We used the simplest ECMWF partitioning that only separates the wave field into wind-sea and swell based on the wave age of each wave component, also accounting for the components direction relative to the wind."*

*Line 285: adjust "artefacts … was".*

**Our response:** Corrected to *"artefacts [...] were"*

*Line 299: under "single" probably "simple" or "straightforward" or "universal" was meant.*

**Our response:** Thank you. What we meant was *"universal"* and this has been corrected to the manuscript.

*Line 312: also the difference between wind and wave direction was taken into account.*

**Our response:** The sentence has been modified to:

*"The partitioning scheme used a typical criteria based on the wave phase speed and wave direction relative to the wind (see Eq. 2)."*